# Co-production of an online research and resource platform for improving the health of young people—The hype project

Cerisse Gunasinghe[1,2]* , Nicol Bergou[1], Shirlee MacCrimmon[1], Rebecca Rhead[1], Charlotte Woodhead[1,3], Jessica D. Jones Nielsen[2], Stephani L. Hatch[1,3]

1 Department of Psychological Medicine, Institute of Psychiatry, Psychology and Neuroscience, King's College London, London, United Kingdom, 2 Department of Psychology, School of Health & Psychological Sciences, City, University of London, London, United Kingdom, 3 Economic and Social Research Council (ESRC) Centre for Society and Mental Health, King's College London, London, United Kingdom

☉ These authors contributed equally to this work.
* cerisse.gunasinghe@city.ac.uk

**Data Availability Statement:** The datasets used and/or analysed during the current study are available with permission from the study PI and

## Abstract

Mental health conditions tend to go unrecognised and untreated in adolescence, and therefore it is crucial to improve the health and social outcomes for these individuals through age and culturally appropriate interventions. This paper aims firstly to describe the development and implementation of the HYPE project platform (a research and resource platform co-designed and co-produced with young people). The second aim is to describe the characteristics of participants who engaged with the platform and an embedded pilot online survey. Participatory action research approach was used to address objectives of the HYPE project. Data were analysed to: (1) help improve access to health and social services, (2) guide provision of information of online resources and (3) deliver complementary community-based events/activities to promote mental health and to ultimately prevent mental health issues. Pilot and main phases of the HYPE project demonstrated the capacity and feasibility for such a platform to reach local, national, and international populations. Analyses demonstrated that the platform was particularly relevant for young females with pre-existing health difficulties. Some of the barriers to involving young people in research and help-seeking are discussed.

## Introduction

The transition period from late childhood to early adulthood is a particularly critical period of development where the early age of onset of mental health conditions (e.g., depression, anxiety, and eating disorders) along with the prevalence, chronicity and comorbidity with long-term persistent and/or recurrent physical health conditions can set the trajectory for an individual's physical and mental health in adult life [1–9]. Further, as many as one in ten 10-24-year-olds have a disability that affects their ability to do daily activities [8]. Findings from longitudinal cohort studies have consistently shown associations between childhood mental

Research Ethics committee by email: rec@kcl.ac.uk on reasonable request. Access to our dataset would require adherence to the HYPE and KCL data transfer and publication policies.

**Funding:** This paper represents independent research funded by the National Institute of Health Research (NIHR) Maudsley Biomedical Research Centre at South London and Maudsley NHS Foundation Trust and King's College London (Grant number BRC-1215–20018). The views expressed are those of the author(s) and not necessarily those of the National Health Service, the NIHR or the Department of Health and Social Care. CG and NB are fully funded by the NIHR Maudsley Biomedical Research Centre at South London and Maudsley NHS Foundation Trust (Grant number BRC-1215–20018). SLH was part-funded by the NIHR Maudsley Biomedical Research Centre at South London and Maudsley NHS Foundation Trust (Grant number BRC-1215–20018) and is supported by the Economic and Social Research Council (ESRC) Centre for Society and Mental Health at King's College London (Grant Reference: ES/S012567/1). SLH also received funding from the Wellcome Trust (Grant Reference: 203380/Z/16/Z) and Impact on Urban Health part of Guy's & St Thomas' Foundation [Grant References: EIC210605 and EIC221208). CW is supported by the ESRC Centre for Society and Mental Health at King's College London (Grant Reference: ES/S012567/1). There was no additional external funding received for this study. The funders had no role in study design, data collection and analysis, decision to publish, or preparation of the manuscript.

**Competing interests:** NO authors have competing interests.

health and poorer mental health in later years [10–12]. Chronic physical health illnesses have a profound impact on the mental well-being of children as young as seven years old [13] and put them at increased risk for developing mental health issues into early adulthood [14]. Research demonstrates the importance of identifying and intervening in comorbid physical and mental health at earlier stages of the life course to prevent negative health and social outcomes.

While social determinants of health are well documented [15–20], including for children and young people in the UK [21–23], those from socioeconomic disadvantaged backgrounds continue to experience poorer health outcomes [24, 25]. For example, there is growing concern about the increase in obesity amongst young people from socially deprived backgrounds, which are associated with other physical health conditions [26] and poor mental health [27]. Regarding other social domains, findings from the South East London Community Health study [28, 29] indicate that compared to older age groups, young people had a greater likelihood for poly substance use [30], being exposed to violence [31] and having the highest proportion of debt [32]. Additionally, exposure to violence and experience of financial debt amongst this group were associated with an increased risk of poor mental health outcomes [31, 32]. Integrated health and social care for young people [32] that is age and culturally appropriate [32, 33] is critically needed to improve mental health outcomes. Woodhead et al (2022) highlight the gaps in research, evidence-base, and service development especially in view of high prevalence and incidence of social welfare legal difficulties [19, 21, 33] Furthermore, there is increasing acknowledgement for such integration with, and across health disciplines given the interface between physical and mental health [34, 35]. However, we need to better understand which social determinants are particularly relevant in late adolescence and early adulthood to tailor resources and improve existing health and social care services and service use.

In response, we developed the HYPE (improving the Health of Young PeoplE) project, described in this paper, which is an online research and resource platform for an ethnically and socioeconomically heterogenous sample of young people (aged 16 years and over). The HYPE project aims to better understand and address the health and social needs of this population through the facilitation of young people's involvement in research; assessment of current social, economic and health experiences of young people; and increase input and access to online and community-based health and social welfare resources and interventions (including clinical research studies). Secondary aims of the project are to pilot, evaluate and assess the feasibility of young people's experience of the HYPE project as well as previous use of health and social care services or programmes. Using this approach, also it will be possible to create a research resource of e-community members who can self-refer or consent to be re-contacted regarding participation in future research studies.

Development and implementation of the HYPE project was guided by the work of van Gemert-Pijnen et al (2011), who proposed five key aspects in optimising the engagement of the target population in online/web-based health interventions and resources: (1) a holistic (multidisciplinary) framework for the development; (2) delivery and uptake of e-health technologies which included development informed by multidisciplinary professions; (3) target users and other key stakeholders; (4) regular review and evaluation to ensure the product meets the needs of the users; consideration of barriers to implementation and how the technology might have implications for the organisation of health care; (5) the use of persuasive strategies to facilitate self-management and finally the use of mixed-method analysis of impact and e-health usage [36]. Furthermore, the Conceptual Framework of Access to Healthcare as proposed by Levesque et al. (2013) offers some important insights into indicators that improve health service use [37, 38]. For young people accessing and engaging with health and social welfare services, the culture of care has been largely insensitive to this group. In a scoping

review, Woodhead et al (2022) highlighted studies that showed low rates of help-seeking amongst young people experiencing social welfare difficulties, though they more frequently reported wishing they had sought advice [35]. This raises significant concern as such support and resources are necessary to help alleviate mental and physical health difficulties and help prevent relapse in long-term conditions [35]. Obstacles to accessing support are particularly pertinent for historically marginalised and socioeconomically disadvantaged young people who are most at risk of health inequalities [39].

This project was proposed as it is widely known that internet use has dramatically increased over recent years, particularly the access and contribution to social media platforms by young people, [40, 41]. In addition, there has been a dramatic increase in the development and use of online/e-health interventions in an attempt to improving accessibility and self-management of various health conditions. Some examples of online web-based resources include, Ieso Digital Health (http://uk.iesohealth.com/), Kooth (https://www.kooth.com/), Shout (https://giveusashout.org/), Headspace (https://www.headspace.com/), Drinkaware (https://www.drinkaware.co.uk/), My Plate (https://www.myplate.gov/) and even Tiktok (https://www.tiktok.com/. Therefore, digital technology and e-health interventions have potential to address the issues of young people's lower rates of health and social welfare service use. However, researchers have highlighted the need to further develop digital health interventions for young people that are sensitive and culturally specific to the needs of this group [42, 43]. There is sparse up-to-date peer-reviewed evidence that illustrates how young people have been involved in the co-creating or modifying such resources to ensure that these meet the needs of this population as well examining how young people engage with health promotion and education via such methods. This is vital given the complex health and social welfare needs of this population as well gaining insight into how to improve access to resources for this population [35].

This protocol paper aims to describe the development and implementation of the HYPE project platform. Our intention here is to transparently document the processes and progress throughout the HYPE project to help others see the benefits of embedding participatory and co-design approaches when conducting similar work. The second aim is to describe the characteristics of participants who engaged with this type of online pilot research survey and associated resource platform.

## Materials and methods

### Design

To address key aims and objectives, the HYPE project, co-designed with the HYPE project stakeholder and advisory groups, is a web-based research and resource platform for young people (aged 16 years and over) living in England. The platform provides a health and social welfare resource directory containing information about online and community-based health and social welfare resources as well as interventions (including clinical trials). The platform also hosts an online pilot survey to assess the health as well as formal (e.g., National Health Service (NHS)) and informal (e.g., community organisation) health service use and support for those who consent to participate.

As recommended by van Gemert-Pijnen et al (2011) [36], mixed-methods data from in-person advisory group meetings, online focus groups (data reported elsewhere) and repeated measures surveys supported clinically informed decisions to guide resource development. Analysis of these data served to; help improve access to health and social services, guide provision of information of online resources and deliver complementary community-based events/activities to promote health and well-being of the target population. Fig 1 illustrates key elements of the HYPE project that are being presented in this paper.

**Public and Patient Engagement and Involvement**

Stakeholder and advisory group consultation

Protocol & platform development

* Website design & online survey tool

* Risk protocols

* Identification of free health and social welfare services (directory)

* Identification of free online and community-based health and social welfare resources and interventions

* Co-creation of social media campaign for Facebook™, Instagram™ and Twitter™

**Development and pilot of research and resource platform**

Ethical approval and project oversight

Release of the HYPE project platform

* Recruitment to online survey

* Access to health and social welfare directory

* Access to online and community-based health and social welfare resources and interventions (including clinical research studies).

**Main phase of HYPE project**

Amended survey protocol

Widening participation & modification of recruitment strategies

Amended recruitment and engagement activities due to COVID-19 in March 2020.

Analyses of research and resource platform use

**Fig 1. Diagram illustrating the stages of development and implementation of the HYPE project platform.**

To address the primary aim of this paper, below is a description of research activities including vital public and patient engagement and involvement that enabled the development and implementation of the HYPE project platform.

### Ethical approval

The HYPE project received ethical approval from the King's College London (KCL) Psychiatry, Nursing and Midwifery Research Ethics Committee for non-clinical research populations (Reference Number:HR-17/18-7535). The research ethics committee approved consenting processes for individuals 16–18 years (with capacity to consent). Those who did not have capacity to consent were not able to provide data for research purposes.

### Setting

The HYPE project research team are based at the Institute of Psychiatry, Psychology & Neuroscience (IoPPN), KCL, (London, UK). London is notable for being the home of one of the largest ethnically and racially diverse communities than other parts of England [44]. The London borough where the IoPPN is based, is characterised by a high population density and higher socio-economic deprivation than the country's average [28]. The location of the study enables an established partnership between the IoPPN and local mental and physical health services

(King's Health Partners) as well as strong established links to community organisations through the Health Inequalities Research Network (HERON). HERON is a research and public engagement network comprising community members and organisations, researchers and healthcare practitioners. Focussing on mental health and the interface between mental and physical health, HERON aims to raise critical awareness of, help people share experiences about, and identify ways to reduce inequalities in health and healthcare. This enables acquired knowledge through this project to have translational value as well as direct benefit for patients and the public.

## Public and patient engagement and involvement

**Establishing and membership of the HYPE stakeholder and advisory groups.** Six months prior to the platform launch (see Fig 2), we carried out 50 in-person consultations with multidisciplinary specialist clinical academics, healthcare providers and intervention specialists (the HYPE stakeholder group), working with young people and/or digital health interventions. Individuals and organisations were either known to, or identified by, the research team [36–38, 42]. These discussions guided the website design, survey content and creation of the digital health and social welfare directory which facilitated a holistic (multidisciplinary) framework for the development of the platform and its content, informed by multidisciplinary professions and allowed for targeting of users and other key stakeholders [36–38, 42].

**Month -6 (pre-pilot)**

**May 2018 - November 2018**

**Establishing stakeholder & young people engagement & involvement**

Multiple in person consultations with specialist clinical academics, healthcare providers and intervention specialists guided the survey and resource development.

Formation of the HYPE project advisory group with diverse young people in relation to age, gender, ethnicity and race.

**Months -6 (pre-pilot)**

**User-testing & consultation**

The HYPE project advisory group were invited to two user testing sessions.

In person consultation with local young people's mental health advisory group.

Feedback resulted in revisions to study name, website design, study procedures, information provided to potential survey participants and the health and social welfare resource directory.

**Months 0-6**

**May 2019**

**User-testing & consultation - during pilot**

Two further in-person user testing sessions. Feedback provided by participants was summarised into recommendations for further improvements of the project and implemented ahead of the main phase of the project which commenced in October 2019.

**Month 15 & 17**

**February 2020 & April 2020**

**User-testing & consultation - main phase**

We held an online focus group with HYPE project advisory group and provided a progress update.

Online focus groups with HYPE community members as well as individuals who expressed interest in taking part in the group via email or a social media campaign.

COVID-19 specific resources, 'Things to do while at home', information relating to the Black Lives Matter movement and resources for LGBTQ+ young people were created as separate additional features of the HYPE project platform.

Regular audit of platform users and suvery responses.

**Fig 2. Flow chart illustrating process of public and patient engagement and involvement.**

In addition, the HYPE project advisory group was formed which included co-authors (CG, NB, SM, RR, CW, JJN & SLH), the HYPE project research team, school/college and undergraduate students, health service users and young people who had previously attended HERON engagement and outreach activities. This advisory group was diverse in relation to age, gender, ethnicity, and race. HYPE project advisory group members were reimbursed with a £15 shopping voucher (in addition to reimbursement of travel expenses if applicable) for their time and contribution [50].

**User-testing and consultation prior to piloting the HYPE project platform.** Following on from establishing the HYPE stakeholder and advisory groups, over two in-person user-testing sessions, the HYPE project advisory group members provided feedback and made suggestions about intended project procedures.

The research team also met with another local young people's mental health advisory group associated with a local health service to discuss the final draft of the research survey and how we might increase engagement with all aspects of the HYPE project platform. Responses from stakeholder and advisory groups were collated and changes to the project and platform were implemented prior to the platform launch and recruitment.

**User-testing and consultation during pilot and main phase of the HYPE project.** After a six-month pilot phase, we held two further user testing sessions in May 2019. We invited the HYPE advisory group as well as HYPE project participants (the HYPE community) who had consented to be recontacted to take part in an initial evaluation of the project progress to date. In February 2020 we held an online focus group with the HYPE project advisory group. This gave young people who were not able to travel to an in-person meeting, an opportunity to be involved and contribute to the project development.

In April 2020, following the onset of the COVID-19 pandemic, we held online focus groups (in compliance with King's College London research ethics and national lockdown measures) with HYPE community members/participants as well as individuals who expressed interest in taking part in the group via email or a social media campaign. Feedback helped guide development of a follow-up survey to understand the needs of young people during the pandemic and to help tailor resources to support them. Suggested resources were added to the health and social welfare resource directory listed on the HYPE project website and were updated and refined based on survey responses. COVID-19 specific resources, 'Things to do while at home', information relating to the Black Lives Matter movement and resources for LGBTQ + young people were created as separate additional features of the HYPE project platform. Fig 2 illustrates the stages and frequency of public and patient engagement and involvement undertaken by the research team (CG and NB) and cofacilitated with members. See Fig 2 for timings of advisory group meetings and Fig 3 for numbers of advisory group meetings and participants at multiple stages of the project. Due to anonymity and confidentiality, we were unable to identify whether advisory and stakeholder members consented and provided responses to the repeated measures online survey.

These latter stages of public and patient engagement and involvement allowed for regular review and evaluation of the HYPE platform, to ensure the it met the needs of the users; considered barriers to implementation and how the technology might have implications for the organisation of health care and (where appropriate) use of strategies to facilitate self-management, as suggested by van Gemert-Pijnen et al (2011) [36].

## Reflexivity

It was important for the research team to spend time ensuring broad outreach and relationship building with organisations and young people themselves. Additionally, being mindful of

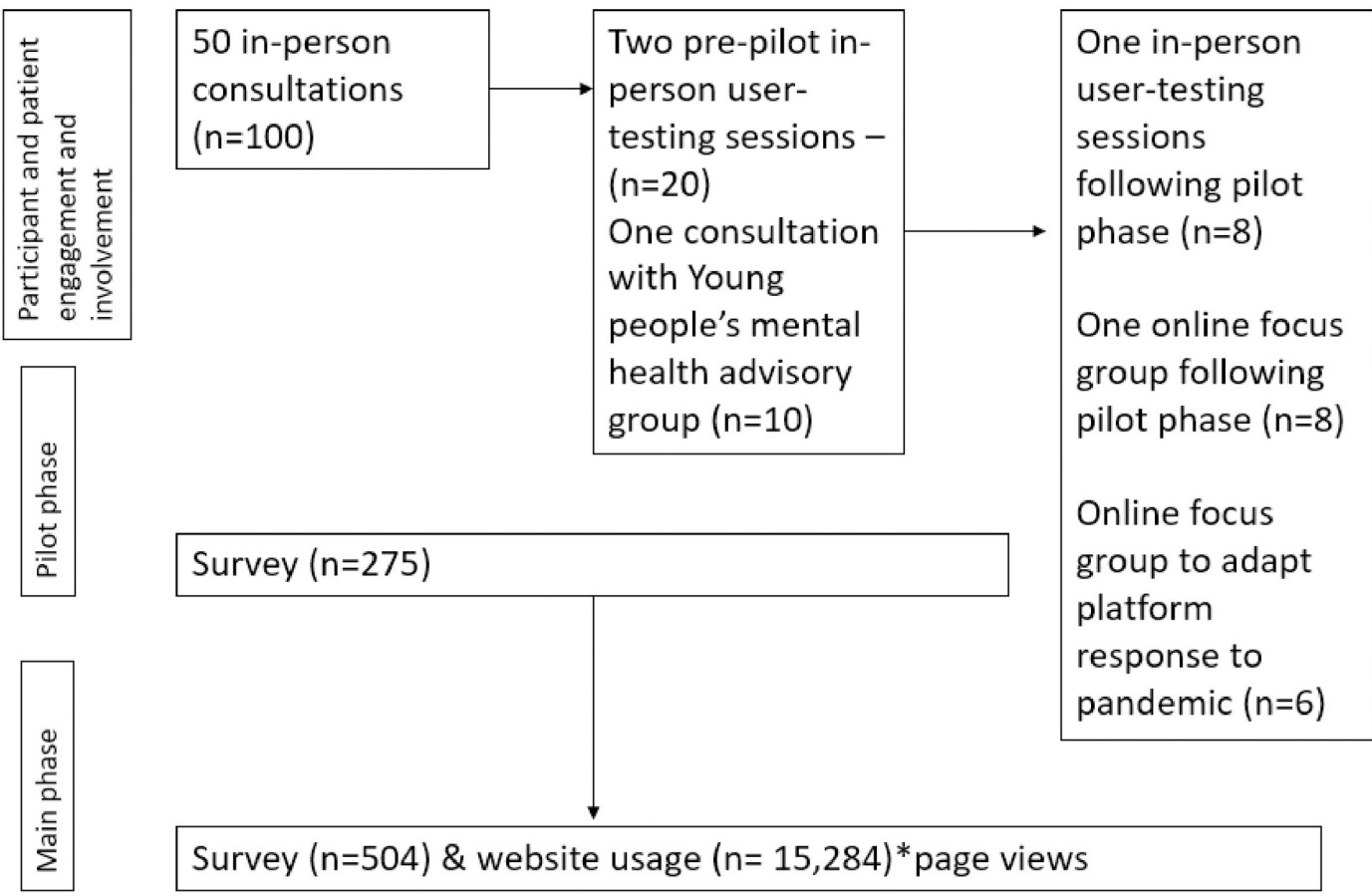

**Fig 3. Flow chart illustrating the stages of research, the methods and the number of participants that took part in the different stages of the project.**

being perceived a positioned as holding power, privilege and status the research team were open and transparent about gaps in their knowledge and experience and invited stakeholders and advisory group members to take leadership roles in discussions and engagement activities. Given the diversity of the research team across multiple personal and social statuses, it was hoped that this would help reduce barriers, bridge the gap and create more equitable working with potential participants.

The following sections describe the research methods, procedure, processes and activities that the research team completed in order to facilitate the survey component of the project. The research survey was one of the parts of the platform developed to address the secondary aim of assessing the health as well as formal (e.g., National Health Service (NHS)) and informal (e.g., community organisation) health service use and support for those who consent to participate.

## Participants

This section provides an overview of the recruitment strategies implemented to invite young people (16 years and over) to: (1) access the platform, (2) participate in the online pilot survey and, (3) take part in activities (e.g., ways young people could get involved with the project, community-based programmes, events) [36–38]. Following on from this, survey sample and inclusion criteria are detailed.

## Recruitment

**Pilot phase—Release of the HYPE project platform.** The pilot phase of the HYPE project platform (https://hypekcl.com/) was launched in November 2018 following preliminary public and patient engagement and involvement (as outlined below). We undertook an iterative process of initially targeting local gatekeepers in community organisations working with young people and a purposively selected group of academic institutions in London, based on existing partnerships or networks including our own HERON Network. In addition, we regularly updated a social media recruitment campaign which was co-created by the research team and members of the HYPE advisory group. These strategies were in parallel with project adverts being placed within the community (e.g., barber shops, nail bars, community centres) and charitable organisations that young people frequented as well as physical and mental health service users. Those who consented to be re-contacted, were invited to take part in follow-up surveys and/or one-to-one face-to-face or telephone interviews to facilitate platform development and further ethically approved young people research.

**Main phase of the HYPE project following revisions to the HYPE project platform.** Following audit and evaluation of the pilot phase with the aim of collating survey responses from a more diverse and representative sample, recruitment strategies were modified for the main phase of the project (October 2019—January 2021). These strategies included linking with racial and ethnic minority social media influencers together with advertising in a local newspaper, religious places of worship, gyms and sport centres/events. In addition, design of social media adverts was tailored to young men and people from racial and ethnic minority backgrounds to encourage the ethos of our approach, "Take part and have your voices heard". Recruitment methods were also scaled up nationally which included study adverts being disseminated by McPin Foundation (a mental health research charity) and MQ (an organisation supporting mental health research). Following the onset of the pandemic, recruitment strategies were largely restricted to online or remote activities. The research team delivered remote presentations to young people service user groups, created a video advertisement for YouTube™ and continued with social media campaigns, which were further impacted by the implementation of stricter guidelines for advertising via social media platforms. Placement of printed project adverts at community centres and in person meetings, engagement and outreach activities were suspended to comply with social distancing and lockdown measures following the outbreak of COVID-19 in March 2020.

## Survey sample and inclusion criteria

A convenience sample who consented to take part and provided responses to the online survey. Participants were eligible if they self-identified as a young person (aged 16 years and over) and residing in the UK.

## Materials

**Development of a pilot online survey.** The HYPE project advisory group and stakeholder group developed topics for the online survey through a process of consensus building. These topics for the survey included: (1) socio-demographics and migration; (2) education (3) mental health symptoms, physical health symptoms and long-standing health conditions; (4) socio-economic status and work-related behaviour; (5) psychosocial factors (e.g., adverse life events, social support, self-esteem, discrimination, neighbourhood characteristics); (6) addictions and substance use; and (7) health service use.

At follow-up (6 and 12 months), questions were repeated to assess physical and mental health outcomes; psychosocial factors (e.g., adverse life events, social networking, self-esteem,

physical activity) and coping strategies. Follow-up surveys also included evaluation (data and analysis outside the scope of this paper) items relating to participants' experience of using the HYPE project platform and use of health services. The relevant mental health and physical health measures for the scope of this paper are outlined below. A full list of measures included in the survey can be accessed from the corresponding author.

**Health outcomes.**   Presence of depression and anxiety symptoms in the most recent two weeks, were assessed using the well-validated *Patient Health Questionnaire (PHQ-9)* [45] and *Generalised Anxiety Disorder-7* (GAD-7) [46, 47]. Both measures are commonly used in primary and secondary care to detect criterion symptoms for depression and/or anxiety and generalised anxiety disorder respectively. Moderate to severe depression or anxiety were indicated where total scores are 10 or above.

*Patient Health Questionnaire-15*. Participants self-reported current severity of 15 somatic experiences that are understood to be among the most common somatic symptoms by answering "During the past 4 weeks (28 days), how much have you been bothered by any of the following problems?" Total scores of 10 or above indicate moderate-severe symptom severity [48].

*Long-standing illness*. Self-reported long-standing health problems, illness or disability by answering Yes or No) to the question "Do you have any long-standing health problems, illness or disability?"

## Procedure

Study adverts and presentation materials invited potential participants to visit the HYPE project platform (https://hypekcl.com/) where they could read a summary of the research project objectives on the home page. If they were interested in taking part in the online research survey, they were asked to click the 'Sign-up' button. The individual would then be directed to the sign-up page where a hyperlink to the participant information sheet (PIS) and easy read PIS were provided, and they were asked to read the PIS and enter their contact email address if still interested in taking part. On receipt of an automated alert email, a member of the research team would email the potential participant with a hyperlink to the PIS with a reminder to re-read the information attached (if not already done so), and a unique link where their electronic consent form and their survey responses were recorded. The consent form and survey were accessed via Qualtrics$^©$ (a web-based survey tool). At follow-up (at 6 and 12 months), similarly a unique survey link was sent to participants who agreed to be recontacted.

## Project oversight

To implement revisions and update the HYPE project platform where appropriate, the research team reviewed progress of the project and obstacles to the delivery in weekly team meetings which aligned with van Gemert-Pijnen et al's (2011) key aspects in optimising the engagement of the target population in online/web-based health interventions and resources [36].

## Development of risk protocols

At the start of the project, the research team discussed possible ways to support to young people who reported self-harm or suicidal intent when completing the Patient Health Questionnaire 9-item depression scale [45] as part of the online survey and developed a risk protocol. In line with ethical considerations foremost and the project aims to increase access and reduce barriers to resources, it was decided that rapid response to participants' survey response (if greater than 0) was important and that our duty of care was to signpost the individual to

appropriate emergency services. We also wanted to test the feasibility of offering and providing an opportunity to speak with a clinical member of the research team to ensure the safety of the participant. If the participant requested to speak to a clinical member of the research team, a mutually convenient time and mode of contact was arranged with the participant. Participants were also able to use the HYPE Project website to request contact (email/telephone call) from a member of the research team if they required further support.

## Amendments to survey protocol

Based on the HYPE Project advisory group feedback following the pilot phase, a monthly prize draw to receive one of three £10 shopping vouchers was introduced into the main study for participants who completed the survey. In addition, postal code was added to both the baseline and follow-up survey to enable the research team to identify locally relevant data on health and social need, as well as to inform the development and tailoring of our online and community-based resources. By providing the first section of a person's postcode, we were able to identify districts and areas where participants were living without compromising their privacy by specifying potential street names or numbers. We also added an item asking the participant where they received information about the project which enabled us to understand which recruitment activities were most effective which would help inform future research project development and guide other researchers in their work.

## Analyses

**Quantitative data.** Completion of the survey was limited to residents of UK to comply with terms of funding and those over 16 years due to the scope of the project, despite the online platform/website being available to international users of all ages. To address the second aim of this paper, Google Analytics supported analysis to identify who engaged with this type of survey and resource platform. Google Analytics does not track users under 18 years old, thus they are excluded from these analyses. In addition, descriptive analyses were conducted using survey data to examine the characteristics of young people who completed the online survey during the pilot/main-project launch. Unweighted frequencies and percentages were calculated for participant socio-demographic characteristics (age, gender, ethnicity, country of birth, highest qualification, employment, and benefit status) and health measures (PHQ-9, GAD-7, PHQ-15, and presence of long-standing illness). Analyses were conducted using STATA 15 [49].

## Results

### Platform users

According to Google Analytics, during the main phase of the study (October 2019 –June 2020) there were almost 9,000 visits to sections of the HYPE project online platform. As illustrated in the graph (see Fig 4 for snapshot of audit conducted in January 2020), spikes in accessing the platform where following targeted engagement strategies via social media campaigns and adaptations to the platform based on feedback received from the HYPE Project's advisory group members. There were 15,284 page views in total. The majority (60%, n = 3964) of users were in the UK, followed by the USA (19%, n = 1274). The two most used web browsers were Chrome (43%) and Safari (31%). The most common mobile phone operating systems used was iOS (61%) followed by Android (38%).

**Survey participants.** Of those who expressed interest in taking part by providing their email address on the (n = 1223), 540 (44/.1%) young people participated in the survey. Within

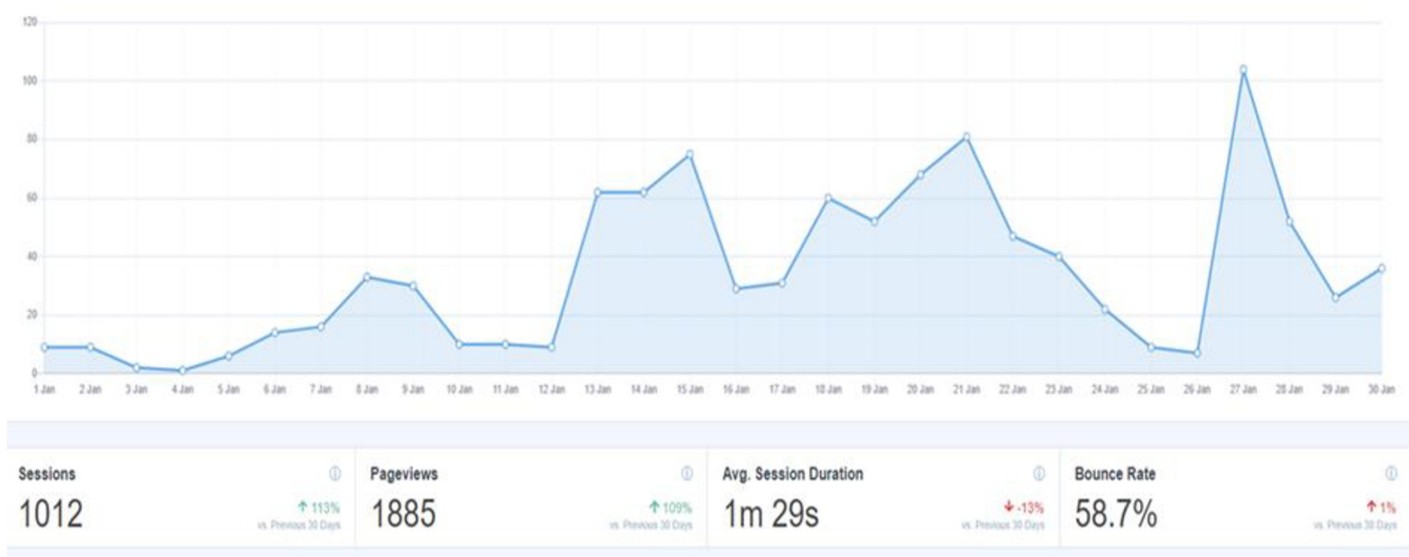

**Fig 4. Graph illustrating website users during January 2020.**

the target population of 16–24-year-olds (n = 433), most of the sample were female (86.8%, n = 376), aged 16–17 years (65.4%, n = 283) and White British (63.3%, n = 274). 272 (62.8%) reported being unemployed and 123 (28.4%) participants were receiving benefits. Within this sample population, as shown in Table 1, 55.7% (n = 241) met criteria for moderate to severe depression, 60.1% (260) for moderate-severe somatic symptoms, 47.6% (206) for moderate-severe anxiety and 70.0% (303) had at least one long-standing illness.

## Discussion

The first aim of this paper was to describe why and how the HYPE project was created as an online research and resource platform for young adults aged 16 years and older. It is well documented that without adequate support or intervention, health and social adversity during mid-late adolescence and early adulthood can have lasting and long-term harmful consequences for individuals [2, 3, 4, 17]. Existing frameworks for optimising the engagement of the target population in online/web-based health interventions and resources [36] and enabling patient-centred access to health care [38] were evidenced in the development of the HYPE project platform which provided an opportunity for young people to participate in research as well as multidisciplinary online and community-based health and social welfare resources and interventions [34]. As suggested by van Gemert-Pijnen et al (2011) [36], implementing a co-design and co-production framework with multidisciplinary stakeholders and young people alongside mixed-methods data (user-testing focus groups and surveys) assisted in gaining better insight and knowledge of the challenges young people are currently facing. Hence, platform users were signposted to resources based on survey responses as well as those that the advisory and stakeholder group identified as relevant to the target population.

The second aim of presenting this protocol was to describe the characteristics of the convenience survey sample. For young people who completed the survey, those experiencing socioeconomic disadvantage (as measured by employment status and benefit receipt) also reported physical and mental health difficulties. Participants had higher rates of mental and physical health conditions, assessed through standardised self-report measures, compared to other

**Table 1. Sample characteristics and proportions of those aged 16–24 years and residing in England who completed the pilot online survey (N = 433).**

| | (N = 433) | |
|---|---|---|
| | **n** | **%** |
| **Gender** | | |
| Male | 44 | 10.2 |
| Female | 376 | 86.8 |
| Other | 13 | 3.0 |
| **Sexual orientation** | | |
| Heterosexual | 241 | 55.7 |
| Gay/Lesbian | 33 | 7.6 |
| Bisexual | 117 | 27.0 |
| Other | 22 | 5.1 |
| Prefer not to say | 20 | 4.6 |
| **Age** | | |
| Under 18 | 283 | 65.4 |
| 18–24 | 150 | 34.6 |
| **Ethnicity** | | |
| White British | 274 | 63.3 |
| White Other | 31 | 7.2 |
| Black African/Black Caribbean | 38 | 8.8 |
| Asian | 58 | 13.4 |
| Mixed/Other | 32 | 7.3 |
| **Country of birth** | | |
| Not Born in UK | 55 | 12.7 |
| Born in UK | 378 | 87.3 |
| **Education level** | | |
| No qualification/Less than GCSE | 36 | 8.3 |
| GCSE or equivalent | 240 | 55.4 |
| A level | 111 | 25.6 |
| Degree or above | 45 | 10.4 |
| **Employment status** | | |
| Unemployed | 272 | 62.8 |
| Employed | 161 | 37.2 |
| **Benefit receipt** | | |
| No benefit received | 310 | 71.6 |
| Benefit received | 123 | 28.4 |
| **Health** | | |
| PHQ-9[a] | 241 | 55.7 |
| PHQ-15[b] | 260 | 60.1 |
| GAD-7[c] | 206 | 47.6 |
| Longstanding illness[d] | 303 | 70.0 |

* Frequencies are unweighted and may not add up due to missing values.

[a] Participants with total scores of 10 and above on PHQ-9 indicating moderate to severe depression.

[b] Participants with total scores of 10 and above on GAD-7 indicating moderate to severe anxiety.

[c] Participants with total score of 10 and above on PHQ-15 indicating moderate to severe level of severity.

[d] Participants with a longstanding illness.

larger cohort studies of children and young people within this age group [5, 7, 8, 10, 13]. These data helped to regularly evaluate the acceptability and appropriateness of the health and social welfare directory which also aligned with the recommendations of van Gemert-Pijnen et al (2011) and Levesque et al. (2013) [36–38], ensuring that multidisciplinary and holistic resources were free to access was intended to reduce financial barriers to accessing services and support [37].

Despite the assumption that young people being perceived as a 'hard to reach' population, there has been ongoing active and invaluable involvement of young people in the project from its inception. Moreover, advisory group members and other young people challenged this notion in episodes of the Beyond the HYPE podcast series (https://hypekcl.com/podcast/). Offering young people various ways to share their experiences contributes to a better understanding why mistrust, stigma, upholding cultural and/or religious norms as well as frustration with being not heard, were some of the barriers to engagement. This was in relation to the platform as well as research and health services (particularly mental health support) more broadly [8, 34]. Previous work from our Health Inequalities Research Group [50] has demonstrated the importance and value of collaborative and coproduction approaches that serve to empower, promote well-being and improve social justice with and for young people.

## Methodological considerations

Due to lessons learned during the pilot phase of the study, subsequent engagement, and recruitment activities considerably increased participation from racial and ethnic minority groups. We have made strides in our understanding of inclusive practices and how to improve engagement with these groups of individuals [34]. For example, holding a continued presence through activities and social media and preparedness to shift priorities of the project and associated tasks. However, there remained low engagement by young men, particularly completion of the survey, despite involving and collaborating with male influencers and organisations specifically working with this population. For example, a podcast episode was co-created titled 'Men's Mental Health with 90s Baby Show' and the research team attended a community event whereby the organisers invited attendees to take part in the HYPE project and visit the platform.

During the main phase of the project, it was ethically permitted to conduct analysis of the online platform/website access. This showed that there was considerably greater engagement with the health and social welfare resource directory, events, and activities, compared to the online research survey. The research team were also able to improve the user journey relating to consenting to, and completing the online survey, which may have increased the conversion rate from initial sign-up to consenting to participate in the survey. However, due to the appropriate age restrictions enforced by Google Analytics, analyses of platform users under 18 years were not captured throughout the project and therefore limited targeted engagement and adaptations to the platform for this population.

A minority of participants who reported mild to severe thoughts of self-harm or suicide, requested a call or further email contact with a study clinician. It is acknowledged that this may have been an additional time and financial cost with a larger sample, however it was considered an additional strength of the project to help reduce barriers to seeking support for poor mental health as well as being able to assist relapse management for those who had preexisting mental health conditions.

Survey fatigue experienced population is likely to have resulted in lower completion rates as it would appear the need for health and social care resources outweighs desires to contribute to *yet another survey*. For those with interest or conducting work in this area, it is important to

consider where there is a duplication of data being collected and where existing data resources can be utilised (for example the Key Data series published by the Association for Young People's Health). Another possible reason for low survey completion rates could be attributed to the positionality of the research team and its affiliation to formal health services. While on one hand an academic institution maybe perceived as a source of information, potential survey participants may have experienced fear and/or mistrust in providing personal information and the implications that may have.

## Strengths and limitations

Facilitating reciprocal relationships was central to this project and of the wider research group whereby it places young people and the community at the centre. Analyses of mixed-method data has illustrated that this digital/web-based platform has potential to reach many young people. The platform in parallel with in-person and community engagement and involvement activities/resources, enabled the research team to attend to both the needs of the individuals and improve access to resources for the wider target population. However, analyses suggests that this type of platform and associated recruitment strategies were more likely to engage individuals with existing experiences of mental and physical health symptoms and/or conditions.

Furthermore, following the onset of the pandemic and national lockdown in the UK in March 2020, the HYPE Project platform was able to rapidly collect data to inform resources on the platform to provide ongoing support for young people. For example, having exams cancelled (national curriculum and those required for higher education institution entrance), disruptions to/uncertainty around starting higher education, less stable romantic relationships, less stable employment, being confined with family during a period in which autonomy can be important), as well as other challenges potentially shared by other age groups (e.g., lack of access to green space, overcrowding, loss of household income, interpersonal conflict, and food insecurity). While we were able to adapt the health and social welfare resource directory to reflect the need to be able to remotely access support and activities, it did mean that our in-person outreach and community activities had to be suspended which may have resulted later disengagement.

## Implications and future directions

Many young people value a sense of agency over their health and wellness. The HYPE project platform endeavoured to inform users about multidisciplinary ways to maintain good health and empower them to make decisions about their help-seeking. With continued evaluation and adaptation, the platform has the potential to support young people experiencing early signs and symptoms of poor health, those in need of (and awaiting) formal health and/or social welfare intervention and supporting relapse prevention in those with existing health conditions.

Future work aims to utilise existing data to extend the HYPE online-platform to provide resources that promote wellbeing in partnerships with existing and new collaborators. For example, we will co-design a life-skills programme, provide vocational opportunities and tailored resources for underrepresented and marginalised young adults which will be made widely available for HYPE community members and the public. To ensure we continue to hear voices of young people, in-depth qualitative studies designed and conducted by students and early carer researchers are in progress. These are exploring and evaluating experiences of educational, health and social welfare programmes and what barriers, if any, exist in accessing these resources. Finally, further analyses of platform use, and survey (including follow-up)

data might be able to help further understand patterns of accessing support either via the platform or other routes. For example, whether people accessing the platform have accessed support that they might have not got elsewhere or got more easily here, whether the resources via the platform provided a bridge to other types of help-seeking, and whether those use the platform are already/active help-seeking rather than those that are not.

## Acknowledgments

A special thanks to the HYPE project participants, the HYPE project young people's advisory and stakeholder groups, our NIHR Maudsley Biomedical Medical Centre Youth Award students, Anna Simpson, Andrew Boateng, The HYPE project research team, Esther Tolani and Paul McCambridge, who have provided guidance in the development and supported the implementation of this platform.

## Author Contributions

**Conceptualization:** Cerisse Gunasinghe, Charlotte Woodhead, Stephani L. Hatch.

**Data curation:** Cerisse Gunasinghe, Nicol Bergou, Shirlee MacCrimmon, Rebecca Rhead.

**Formal analysis:** Cerisse Gunasinghe, Nicol Bergou, Rebecca Rhead, Stephani L. Hatch.

**Methodology:** Cerisse Gunasinghe, Nicol Bergou, Shirlee MacCrimmon, Rebecca Rhead, Charlotte Woodhead, Jessica D. Jones Nielsen, Stephani L. Hatch.

**Project administration:** Cerisse Gunasinghe, Nicol Bergou.

**Supervision:** Cerisse Gunasinghe, Stephani L. Hatch.

**Visualization:** Stephani L. Hatch.

**Writing – original draft:** Cerisse Gunasinghe.

**Writing – review & editing:** Cerisse Gunasinghe, Nicol Bergou, Shirlee MacCrimmon, Rebecca Rhead, Charlotte Woodhead, Jessica D. Jones Nielsen, Stephani L. Hatch.

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
