## [Decision Letter · Decision Letter 0]

26 Apr 2023

PONE-D-22-29685Co-production of an online research and resource platform for improving the health of young people - The HYPE Project.PLOS ONE

Dear Dr. Gunasinghe,

Thank you for submitting your manuscript to PLOS ONE. After careful consideration, we feel that it has merit but does not fully meet PLOS ONE’s publication criteria as it currently stands. Therefore, we invite you to submit a revised version of the manuscript that addresses the points raised during the review process.

We look forward to receiving your revised manuscript.

Kind regards,

Apurva Kumar Pandya, PhD

Academic Editor

PLOS ONE

Journal Requirements:

“This paper represents independent research funded by the NIHR Maudsley Biomedical Research Centre at South London and Maudsley NHS Foundation Trust and King’s College London. The views expressed are those of the author(s) and not necessarily those of the NHS, the NIHR or the Department of Health and Social Care. SH is part-funded by the NIHR Maudsley Biomedical Research Centre at South London and Maudsley NHS Foundation Trust and supported by the Economic and Social Research Council (ESRC) Centre for Society and Mental Health at King's College London (ESRC Reference: ES/S012567/1). SH also receives funding from Impact on Urban Health and Wellcome Trust. C.W. is supported by the ESRC Centre for Society and Mental Health at King's College London (ESRC Reference: ES/S012567/1). The funders did not have a role in the study design; collection, analysis or interpretation of data; the writing of the manuscript; or in the decision to submit the manuscript for publication. These data can be accessed through the corresponding author.”

4. We note that Figure 2 in your submission contain [map/satellite] images which may be copyrighted. All PLOS content is published under the Creative Commons Attribution License (CC BY 4.0), which means that the manuscript, images, and Supporting Information files will be freely available online, and any third party is permitted to access, download, copy, distribute, and use these materials in any way, even commercially, with proper attribution. For these reasons, we cannot publish previously copyrighted maps or satellite images created using proprietary data, such as Google software (Google Maps, Street View, and Earth). For more information, see our copyright guidelines: http://journals.plos.org/plosone/s/licenses-and-copyright.

“I request permission for the open-access journal PLOS ONE to publish XXX under the Creative Commons Attribution License (CCAL) CC BY 4.0 (http://creativecommons.org/licenses/by/4.0/). Please be aware that this license allows unrestricted use and distribution, even commercially, by third parties. Please reply and provide explicit written permission to publish XXX under a CC BY license and complete the attached form.

Additional Editor Comments (if provided):

I appreciate the authors for doing this wonderful work; however, the study has major flaws in its presentation. I agree with the reviewers comments. Introduction need to be crisp. The results section is surprisingly very short and seems it does not adequately delivers what authors should. I do not find depth in the analysis. I do not comprehend key themes that came out of the thematic analysis. Further, I fail to find triangulation of analysis of the data from difference data collection methods. Discussion section need to be strengthened in terms of highlighting novelty, similarities and differences of the findings in the context of existing literature. This manuscript requires major revision.

Reviewers' comments:

Reviewer's Responses to Questions

**Comments to the Author**

1. Is the manuscript technically sound, and do the data support the conclusions?

Reviewer #1: Partly

Reviewer #2: Yes

2. Has the statistical analysis been performed appropriately and rigorously? 

Reviewer #1: I Don't Know

Reviewer #2: Yes

3. Have the authors made all data underlying the findings in their manuscript fully available?

Reviewer #1: Yes

Reviewer #2: Yes

4. Is the manuscript presented in an intelligible fashion and written in standard English?

Reviewer #1: Yes

Reviewer #2: Yes

5. Review Comments to the Author

Reviewer #1: INTRODUCTION

The introduction is written well in terms of outlining the need for the HYPE platform, however the paper is much more about the coproduction of the platform through advisory groups and stakeholder engagement and the online nature of the intervention. It would better situate the reader if more evidence was presented at this stage about the different online platforms that are out there for young people, how they have used co-production in their development (or not) and more about ‘the work of van Germert-Pijnen and the proposed five key aspects in optimising the engagement of the target population in online/web-based health interventions and resource. For example we know that TikTok is being used as a search engine by young people. I feel the introduction would benefit from focusing on the way young people engage online. Also you mention that the work was guided by Germert-Pijnen and Levesque, however this is not revisited in the analysis or the discussion. How did these inform the study?

I suggest removing the qualitative study questions and hypothesis as this confuses the purpose of the paper. This whole section created confusion as to what you were presenting.

METHODS:

It is clear that there was lots of engagement and work gone into making the platform relevant for young people which is admirable and well thought through, it is a great programme. However, the section between the introduction and the results is convoluted and would benefit from being considerably reduced with better use of figures/table/graphics to demonstrate many of the steps. I suggest the following:

• A creative figure showing the different elements of the HYPE programme and which are being presented in this paper as I struggled to keep focus on your main aim.

• A diagram or table showing the stages of research, the methods and the number of participants that took part in the different stages. For example, I could not decipher how many people took part in FDGs, how many took place and how often.

• Cut the level of detail provided on the process and just focus on giving the reader an overview of what you did

• A section that describes the membership of the advisory group, including diversity and numbers, drop out rates etc, to give a feel for who was advising the development of the project and how they found it. Also as the title has the word co-production in it, I would expect to see some reflexivity and consideration of power. Are any of the authors from the advisory group? I would expect their input on this if it is really adhering to co-production principles.

Line 63 sentence needs ‘this’ removing. ‘Integrated health and social care for this young people (32) that is age and culturally appropriate (32, 33) is critically needed to improve mental health outcomes.

256 the HYPE project advisory group members provided feedback and made 257 suggestions about intended project procedures.- how many did this?

Line 357 This is quite a big statement, what is it based on - It is estimated that these rates would be doubled or more if data were available for those under 18 years.

Limitation google analytics for under 16s – is this in limitations?

RESULTS

The results section is very short and I am not sure it delivers what you intend. I felt that some of the results were interspersed in the methods and discussion sections. Also I couldn’t find the results from the FDGs or any themes that came out of the thematic analysis. Was it the resources section? This needs much more clarity. Again suggest being clear what methods and data you are presenting and ensure that it links to your key messages and aims.

DISCUSSION

The discussion feels like an extension of the results, suggest better links to the theory and concepts earlier in the introduction and presenting some key take home messages that draw and situate the study in the current literature.

OVERALL COMMENT:

There is a lot of data and processes described and presented in the paper, I would suggest narrowing the focus of this paper to be either about the process or about the survey data/FGDs as currently it is difficult to follow. Alternatively consider a more concise description and a stronger discussion that links to the literature. It would be great to see the discussion linked to a particular framing and other co-production research that uses online platforms. It’s a great project, I look forward to the revisions.

Reviewer #2: Thank you for your submission. It was a good read. I appreciate your well the manuscript was written. Every section had detailed information. It is challenging to engage community in projects so I also liked how much attention was given to the outreach strategies.

6. PLOS authors have the option to publish the peer review history of their article (what does this mean?). If published, this will include your full peer review and any attached files.

Reviewer #1: **Yes: **Kim Ozano

Reviewer #2: No

<quillbot-extension-portal></quillbot-extension-portal>

---

## [Author Response · Author response to Decision Letter 0]

23 Oct 2023

Dear Professor Apurva Kumar Pandyan and reviewers,

Thank you for your letter requesting that we revise and resubmit our manuscript entitled " Co-production of an online research and resource platform for improving the health of young people - The HYPE Project." recently submitted to the PLOS ONE (PONE-D-22-29685).

We are grateful to your reviewers for their helpful suggestions. In the response to reviewers’ document, we have included a reply to their comments and explain how we have revised the paper (below in bold italics). We have provided both clean and highlighted (indicating revisions) versions of our paper as part of our re-submission.

We hope our responses and our revised article are satisfactory. 

Thank you in advance for re-considering this manuscript.

Sincerely, 

Dr Cerisse Gunasinghe

---

## [Decision Letter · Decision Letter 1]

30 Jan 2024

Co-production of an online research and resource platform for improving the health of young people - The HYPE Project.

PONE-D-22-29685R1

Dear Dr.Cerisse Gunasinghe

We’re pleased to inform you that your manuscript has been judged scientifically suitable for publication and will be formally accepted for publication once it meets all outstanding technical requirements.

Kind regards,

Mudassir Khan, Ph.D

Academic Editor

PLOS ONE

Reviewers' comments:

Reviewer's Responses to Questions

**Comments to the Author**

1. If the authors have adequately addressed your comments raised in a previous round of review and you feel that this manuscript is now acceptable for publication, you may indicate that here to bypass the “Comments to the Author” section, enter your conflict of interest statement in the “Confidential to Editor” section, and submit your "Accept" recommendation.

Reviewer #2: All comments have been addressed

Reviewer #3: All comments have been addressed

2. Is the manuscript technically sound, and do the data support the conclusions?

Reviewer #2: Yes

Reviewer #3: Yes

3. Has the statistical analysis been performed appropriately and rigorously? 

Reviewer #2: Yes

Reviewer #3: (No Response)

4. Have the authors made all data underlying the findings in their manuscript fully available?

Reviewer #2: Yes

Reviewer #3: Yes

5. Is the manuscript presented in an intelligible fashion and written in standard English?

Reviewer #2: Yes

Reviewer #3: Yes

6. Review Comments to the Author

Reviewer #2: Thank you for your submission. It was a good read. I appreciate your well the manuscript was written. Every section had detailed information. It is challenging to engage community in projects so I also liked how much attention was given to the outreach strategies.

Reviewer #3: (No Response)

7. PLOS authors have the option to publish the peer review history of their article (what does this mean?). If published, this will include your full peer review and any attached files.

Reviewer #2: **Yes: **Kriti Vashisht

Reviewer #3: No

---

## [Editor Report · Acceptance letter]

6 Mar 2024

PONE-D-22-29685R1 

PLOS ONE

Dear Dr. Gunasinghe, 

I'm pleased to inform you that your manuscript has been deemed suitable for publication in PLOS ONE. Congratulations! Your manuscript is now being handed over to our production team.

Kind regards, 

on behalf of

Dr. Mudassir Khan 

Academic Editor

PLOS ONE